# PRMT7 Inhibitor SGC8158 Enhances Doxorubicin-Induced DNA Damage and Its Cytotoxicity

**DOI:** 10.3390/ijms232012323

**Published:** 2022-10-14

**Authors:** Ahyeon Jeong, Yena Cho, Minkyeong Cho, Gyu-Un Bae, Dae-Geun Song, Su-Nam Kim, Yong Kee Kim

**Affiliations:** 1Muscle Physiome Research Center and Drug Information Research Institute, College of Pharmacy, Sookmyung Women’s University, Seoul 04310, Korea; 2Natural Products Research Institute, KIST Gangneung, Gangneung 25451, Korea; 3Division of Bio-Medical Science and Technology, University of Science and Technology KIST School, Seoul 02792, Korea

**Keywords:** protein arginine methyltransferase 7, SGC8158, DNA damage response, senescence, cell cycle

## Abstract

Protein arginine methyltransferase 7 (PRMT7) regulates various cellular responses, including gene expression, cell migration, stress responses, and stemness. In this study, we investigated the biological role of PRMT7 in cell cycle progression and DNA damage response (DDR) by inhibiting PRMT7 activity with either SGC8158 treatment or its specific siRNA transfection. Suppression of PRMT7 caused cell cycle arrest at the G_1_ phase, resulting from the stabilization and subsequent accumulation of p21 protein. In addition, PRMT7 activity is closely associated with DNA repair pathways, including both homologous recombination and non-homologous end-joining. Interestingly, SGC8158, in combination with doxorubicin, led to a synergistic increase in both DNA damage and cytotoxicity in MCF7 cells. Taken together, our data demonstrate that PRMT7 is a critical modulator of cell growth and DDR, indicating that it is a promising target for cancer treatment.

## 1. Introduction

Protein arginine methyltransferases (PRMTs) are responsible for cellular arginine methylation, and nine PRMTs have been identified in mammals to date [1]. PRMTs produce three types of methylarginine: monomethylarginine (MMA), asymmetric dimethylarginine (ADMA), and symmetric dimethylarginine (SDMA) [2,3]. MMA is produced by all types of PRMTs, but the enzymes that produce ADMA and SDMA are distinct from each other. Type I PRMTs (PRMT1, 2, 3, 4, 6, and 8) generate both MMA and ADMA, whereas type II PRMTs (PRMT5 and 9) produce both MMA and SDMA [4,5]. PRMT7 produces only MMA on its substrate proteins, including histones (H3, H4, H2B, and H2A) and non-histone proteins (DVL3, G3BP2, eIF2α, Hsp70, and β-catenin), and is classified as a type III enzyme [6,7]. PRMT7 has two S-adenosylmethionine (SAM)-binding motifs that are required for its methyltransferase activity [8,9]. PRMT7 overexpression has been closely associated with breast cancer and leukemia [10,11]. In PRMT7-overexpressing cells, PRMT7 regulates epithelial-mesenchymal transition (EMT) by binding to the E-cadherin promoter and repressing its transcription through epigenetic modifications [11]. During EMT, PRMT7 methylates the scaffolding protein SH3 and multiple ankyrin repeat domain 2 (SHANK2) and activates endosomal focal adhesion kinase/cortactin signaling, contributing to cancer cell invasion, metastasis, and malignancy [12]. However, the biological role of PRMT7 is relatively poorly studied compared to other PRMTs because it not only shares substrate proteins with other PRMTs, but also lacks selective inhibitors. Recently, SGC8158 (Figure 1A), an active probe of SGC3027, was reported as a pharmacological inhibitor of PRMT7 [13], which may enable in-depth research to explore the cellular functions of PRMT7.

The DNA damage response (DDR) is a sophisticated cellular network that orchestrates DNA damage checkpoint activation and facilitates the removal of DNA lesions. The DDR network contains a repair mechanism for double-stranded DNA breakage (DSB), which acts as a guardian to maintain intracellular genome stability against exogenous and endogenous DNA damage stresses [14,15]. Aberrant DSB repair processes may lead to genome instability and various human diseases, particularly cancers [16]. Two major pathways are responsible for repairing DSBs: homologous recombination (HR) and non-homologous end-joining (NHEJ). As HR requires a sister chromatid as a template, it is error-free and occurs predominantly during the late S/G_2_ phase of the cell cycle. On the other hand, NHEJ is error-prone and induces genomic instability because it directly connects non-homologous ends without a template strand and occurs throughout the cell cycle [17,18]. It has been well demonstrated that PRMTs are tightly associated with DDR by methylating and regulating many substrates responsible for DSB repair. Several DDR-associated proteins have been identified as PRMT1 substrates, including MRE11 [19], BRCA1 [20], 53BP1 [21], and hnRNPUL1 [22]. In addition, PRMT5 plays a crucial role in regulating genome stability by methylating DDR proteins, including 53BP1 [23], FEN1 [24], RAD9 [25], and RUVBL1 [26]. As such, arginine methylation is indispensable for maintaining genomic stability by tightly regulating DDR; however, the role of PRMT7 remains to be elucidated.

In this study, we investigated the cellular functions of PRMT7 in cell cycle progression and DDR using a PRMT7-specific inhibitor SGC8158 or its specific knockdown system. Abrogation of PRMT7 not only inhibited the proliferation of various human cancer cells but also suppressed DSB repair pathways, both HR and NHEJ, leading to cellular senescence. More importantly, PRMT7 inhibition potentiated doxorubicin-induced DNA damage and increased its cytotoxicity, indicating that this combined therapy might be a promising strategy for cancer treatment.

## 2. Results

### 2.1. PRMT7 Inhibition Causes Growth Inhibition in Various Cancer Cells

To investigate the biological activity of SGC8158, we first confirmed that SGC8158 inhibits PRMT7 activity in cellular systems. As shown in Figure 1B, SGC8158 treatment markedly decreased MMA levels of Hsp70, the best characterized PRMT7 substrate [13], which was consistent with the observation that MMA levels were reduced in PRMT7 knockdown cells. Next, we evaluated the anti-proliferative effects of SGC8158 in various human cancer cell lines. The IC_50_ values of SGC8158 ranged from 2 to 9 μM (Figure 1C). Interestingly, SGC8158 showed similar anti-proliferative effects in KB (IC_50_, 2.0 μM) and multidrug-resistant (MDR) cells KBV20C (IC_50_, 2.2 μM). In addition, the inhibitory effect of SGC8158 in a non-cancer cell mouse embryonic fibroblast (MEF; IC_50_, 28.4 μM) was much weaker than those in other cancer cells. The growth inhibitory effects of SGC8158 were further confirmed by the colony-forming assay. SGC8158 treatment significantly attenuated the colony formation of A549 cells, which was further confirmed in PRMT7 knock-down cells (Figure 1D). Similar effects were observed in MCF7 cells (Appendix A). These results show that SGC8158, a PRMT7 inhibitor, possesses a broad spectrum of anti-proliferative effects on various human cancer cells, as well as MDR cancer cells.

### 2.2. PRMT7 Inhibition Induces Cell Cycle Arrest and Cellular Senescence via p21 Accumulation

To define the possible molecular mechanism of PRMT7 regulation of cell growth, we first performed cell cycle analysis under conditions in which PRMT7 was suppressed. PRMT7 inhibition by SGC8158 treatment resulted in a 14 percentage point increase in the G_1_ phase cell population compared to that in the control group (Figure 2A). Next, we determined the expression levels of cell cycle-related proteins. As shown in Figure 2B, PRMT7 inhibition did not affect the expression of cyclin-dependent kinases (CDKs) and G_1_ cyclins (cyclin D_1_ and cyclin E); however, the expression of endogenous CDK inhibitors (CKIs), including p21, p16, and p27, were dramatically increased, which was accompanied by a decrease in phospho-Rb levels. Concomitantly, mitotic cyclin B_1_ levels were also decreased (Figure 2B). Altogether, these findings suggest that the suppression of PRMT7 results in the accumulation of CKIs and subsequent inhibition of CDK activity, interfering with cell cycle progression. In addition, there were no changes in the levels of the anti-apoptotic molecule Bcl2 and cleavage of PARP, implying that PRMT7 inhibition does not affect apoptotic cell death. Therefore, we determined whether PRMT7 inhibition-induced growth arrest caused cellular senescence. The number of senescence-associated β-galactosidase (SA-β-gal)-positive cells, a hallmark of cellular senescence, was dramatically increased in SGC8158-treated cells compared to the control group (Figure 2C), which was further confirmed by similar effects in PRMT7 knockdown cells (Figure 2D). To determine how PRMT7 inhibition increases p21 levels, we determined the mRNA levels of p21. As shown in Figure 2E, p21 mRNA levels were not altered by SGC8158 treatment or PRMT7 knockdown, suggesting that PRMT7 does not affect the transcription of p21. On the other hand, PRMT7 seemed to affect p21 protein stability, as evidenced by the observation that the level of p21 protein in the PRMT7-suppressed cells in the presence of cycloheximide was maintained longer compared to control cells (Figure 2F,G).

### 2.3. PRMT7 Regulates DDR

To elucidate the biological role of PRMT7 in DDR, DNA damage-induced CHK1 and CHK2 signaling pathways were examined first. Inhibition or depletion of PRMT7 weakly diminished CHK1 phosphorylation (p-Ser345) but did not affect CHK2 phosphorylation (p-Thr68) (Figure 3A,B). However, γH2AX levels were slightly increased by inhibiting PRMT7 (Figure 3A,B), suggesting that PRMT7 might be involved in DNA damage and repair processes. Next, we determined whether PRMT7 is necessary for DNA repair. After inducing DNA damage by treatment with etoposide for 2 h, the cells were incubated for an additional 4 h in fresh medium without etoposide. DNA repair was monitored by measuring γH2AX levels in the cells. As shown in Figure 3C, the γH2AX band induced by etoposide clearly disappeared in the control group after 4 h of release; however, the γH2AX levels in SGC8158-treated cells remained. Additionally, γH2AX levels were maintained even after etoposide release in PRMT7 knock-down cells (Figure 3D). To further support these findings, we performed γH2AX foci analysis using immunofluorescence staining. Consistent with the above results, γH2AX foci in PRMT7-suppressed cells persisted after etoposide release (Figure 3E,F). All these results indicate that PRMT7 is involved in the DNA repair process.

### 2.4. PRMT7 Regulates DNA Repair Processes

Because HR and NHEJ are major pathways for DSB repair, we next investigated the possible role of PRMT7 activity in DSB repair pathways using DR-GFP (for HR) and Ei5-GFP (for NHEJ) reporter assays [23] (Figure 4A). We observed that PRMT7 depletion or inhibition resulted in a significant reduction in both HR and NHEJ repair processes (Figure 4B,C). Since BRCA2 has been demonstrated to be an important regulator of HR [27,28], we examined BRCA2 expression under PRMT7 suppression conditions. As shown in Figure 4D, BRCA2 expression was reduced by the inhibition of PRMT7. Since 53BP1 and the Ku70/80 heterodimer localize to DSB sites to facilitate NHEJ [29,30,31], we next determined their expression levels. PRMT7 inhibition did not affect Ku80 expression but slightly decreased 53BP1 levels. Taken together, our findings suggest that PRMT7 may be associated with DDR by regulating the expression levels of genes related to DSB repair.

### 2.5. PRMT7 Inhibitor SGC8158 with Doxorubicin Potentiates the Cytotoxicity

Given our finding that PRMT7 inhibition does not achieve normal DNA repair, we hypothesized a synergistic effect between SGC8158 and DNA-damaging agents. To investigate the synergistic effects, we used p53 wild-type cancer cells, MCF7 and U2OS, because these cells are more sensitive to DNA-damaging agents. As expected, we found that the combination of SGC8158 and doxorubicin synergistically increased γH2AX foci in MCF7 cells (Figure 5A,B), which was further confirmed by immunoblot analysis showing a marked increase in γH2AX levels under the same conditions (Figure 5C). In parallel, p21 expression was dramatically increased by the combination treatment (Figure 5C). To further investigate the potential synergistic effect, we combined the two drugs at different concentrations. As individual agents, SGC8158 showed an inhibitory effect with 4.16 μM of IC_50_ value, while doxorubicin suppressed cell growth with 2.73 μM of IC_50_ value (Figure 5D). However, treatment with SGC8158 (1 or 3 μM) dramatically lowered the IC_50_ value of doxorubicin (Figure 5E). In addition, in a combinatorial setting, lower concentrations of SGC8158 and doxorubicin inhibited proliferation (Figure 5D,E), suggesting a synergistic effect between these two drugs. To prove the synergistic effect, we further examined the combination index (CI) value using the CompuSyn program. As shown in Figure 5F, the CI score was less than 1, indicating that doxorubicin and SGC8158 had a pharmacological synergistic effect. Furthermore, the Bliss synergy score was calculated using the Bliss independence model in the Synergyfinder. As shown in Figure 5E, we confirmed that there was a synergistic effect between the two drugs because the Bliss synergy score was 9.705. Similar synergistic effect was observed in U2OS cells (Appendix A). Finally, this synergistic effect was further strengthened by the observation that cellular senescence significantly increased in combination of SGC8158 and doxorubicin (Figure 5H). Collectively, our data suggest that SGC8158, a PRMT7 inhibitor, interferes with the DDR of cancer cells in combination with doxorubicin, leading to the potentiation of doxorubicin sensitivity.

## 3. Discussion

As a distinguishing feature from other PRMTs, PRMT7 generates only MMA on its substrate proteins and contains two SAM-binding motifs required for its methyltransferase activity [8,9]. In addition to histones (H2B at R29/R31/R33 and H4 at R17/R19) [9], PRMT7 methylates non-histone substrates including Dvl3 [32], G3BP2 [33], eIF2α [34], Hsp70 [13], and β-catenin [7]. In addition, since PRMT7 overexpression has been implicated in various cancers, including breast cancer and leukemia [10,11], PRMT7 is emerging as a potential target for cancer treatment. Recently, SGC8158, a potent and selective inhibitor of PRMT7, was developed to inhibit the methyltransferase activity of PRMT7 in a SAM-competitive manner [13]. In this study, we further characterized the biological activities of SGC8158 by comparing and analyzing them with those of the PRMT7 knockdown cells. SGC8158 significantly suppressed the proliferation of various cancer cells, including MDR cancer cells, due to cell cycle arrest and cellular senescence via p21 accumulation (Figure 1 and Figure 2). These effects were in good agreement with those observed in PRMT7 knockdown cells, suggesting that PRMT7 governs cell growth by regulating the cell cycle. A previous report showed that the PRMT7-deficient increase in p21 expression appears to be caused by diminished expression of DNA methyltransferase 3b (DNMT3b) [35]; however, we suggest that p21 accumulation may be due to protein stabilization. The molecular mechanisms underlying how PRMT7 regulates p21 turnover remain to be explored. The KBV20C cells, a well-characterized MDR cancer cell line, are derived from KB cells and are resistant to various chemotherapeutic agents, including vincristine, paclitaxel, and doxorubicin [36,37,38]. Since MDR has been considered a major hurdle for successful cancer therapy [39,40], it is noteworthy that SGC8158 is also effective in MDR cancer cells, providing an important clue in developing strategies to overcome MDR.

Although the role of PRMT7 in DDR has not been extensively studied, there are few reports of its involvement in DDR. PRMT7 methylates histone H2AR3 and H4R3 on the promoter regions of DNA repair genes, including *ALKBH5*, *APEX2*, *POLD1*, and *POLD2*, which leads to the transcriptional repression of these genes [41]. Consequently, PRMT7 depletion may confer resistance to chemotherapeutic agents such as cisplatin, chlorambucil, and mitomycin C [41]. In contrast, a previous study reported that PRMT7 knockdown enhanced sensitivity to camptothecin, indicating that it may play a role in DDR [42]. This assumption that PRMT7 regulates DDR is consistent with our observations that either pharmacological inhibition or knockdown of PRMT7 impedes DNA repair processes (HR and NHEJ) (Figure 3 and Figure 4), and HR and NHEJ pathways have been well demonstrated as molecular mechanisms for repairing DSB-induced DNA damage. However, it is difficult to clearly distinguish between the HR and NHEJ pathways because they influence each other closely. Although the exact mechanism of PRMT7 involvement in DDR was not elucidated in this study, it is likely that the regulation of BRCA2 expression is closely involved in DDR. Identifying the DDR substrates of PRMT7 would reveal how PRMT7 specifically modulates DNA repair and sensitivity to DNA damage agents. Interestingly, inhibition of the DSB repair pathway by SGC8158 appeared to synergistically increase the cytotoxicity of the DNA-damaging agent doxorubicin (Figure 5). In addition, it is interesting that SGC8158 is also effective in MDR cancer cells, providing an important clue in establishing strategies to overcome MDR cancer.

In summary, our data demonstrated that specific inhibition of PRMT7 with SGC8158 blocks cell proliferation and DSB repair pathways, both HR and NHEJ, in human cancer cells, leading to cellular senescence. Furthermore, PRMT7 inhibition potentiated doxorubicin-induced DNA damage and increased cytotoxicity. These findings suggest that the combination therapy of a PRMT7 inhibitor with a DNA-damaging agent might provide promising insights into establishing an anti-cancer strategy.

## 4. Materials and Methods

### 4.1. Cell Culture and Transfection

U2OS (human osteosarcoma cells), MCF7 (human breast cancer cells), U87 (human glioblastoma cancer cells), HepG2 (human liver cancer cells), and MEF (mouse embryonic fibroblast) cells were cultured in Dulbecco’s modified Eagle’s medium (DMEM) supplemented with 10% fetal bovine serum (FBS) and 100 U/mL penicillin-streptomycin (Hyclone Laboratories, Inc., Logan, UT, USA). A549 (human non-small lung cancer cells), KB (human oral squamous cell carcinoma cells), and HL-60 (human acute myeloid leukemia cells) were cultured in RPMI-1640 (Roswell Park Memorial Institute-1640) supplemented with 10% FBS and 100 U/mL penicillin-streptomycin (Hyclone Laboratories). All cell lines were obtained from American Type Culture Collection (ATCC; Manassas, VA, USA). Multidrug-resistant KBV20C cells were derived from KB cells and cultured with 20 nM vincristine under growth conditions to preserve MDR characteristics, as described previously [36,37]. All the cells were maintained at 37 °C under 5% CO_2_ in a humidified chamber. For siRNA transfection, cells were transfected with Transit-X2^TM^ (Mirus Bio, Madison, WI, USA) according to the manufacturer’s protocol. All siRNAs were synthesized by Integrated DNA Technologies Pte. Ltd. (Singapore). The following siRNAs were used in this study: human PRMT7-siRNA sequence: 5′-GCUAACCACUUGGAAGAUAAAAUTA-3; human BRCA2-siRNA: 5-CAAGAAGCAUGUCAUGGUAAUACTT-3; and human Ku80-siRNA sequence: 5′-GGAAGUGAUAUAGUUCCUUUCUCTA-3′.

### 4.2. Constructs, Reagents, and Antibodies

SGC8158, doxorubicin, and CHX were obtained from Sigma-Aldrich (St. Louis, MO, USA) and etoposide was obtained from Enzo Life Sciences (Farmingdale, NY, USA). The following antibodies were used for western blotting analysis or immunoprecipitation (IP): β-actin (Santa Cruz Biotechnology, Dallas, TX, USA, sc-47778), p53 (Santa Cruz Biotechnology, sc-126), cyclin B1 (Santa Cruz Biotechnology, sc-752), cyclin D1 (Santa Cruz Biotechnology, sc-8396), cyclin E (Santa Cruz Biotechnology, sc-377100), CDK2 (Santa Cruz Biotechnology, sc-6248), CDK6 (Santa Cruz Biotechnology, sc-7961), PARP1 (Santa Cruz Biotechnology, sc-7150), 53BP1 (Santa Cruz Biotechnology, sc-515841), BRCA2 (Santa Cruz Biotechnology, sc-293185), Hsp70 (Enzo Life Sciences, ADI-SPA-810-F), PRMT7 (Abcam, Cambridge, UK, ab179822, ab181214), MMA (Abcam, ab415), p-Rb (Abcam, ab24), p21 (Cell Signaling Technology, Danvers, MA, USA, #2947), p16 (Cell Signaling Technology, #80772), Rb (Cell Signaling Technology, #9309), CDK (Cell Signaling Technology, #12790), γH2AX (Cell Signaling Technology, #9718), p-chk1 (Cell Signaling Technology, #2348S), p-chk2 (Cell Signaling Technology, #2661S), Ku80 (Cell Signaling Technology, #2180S), p27 (Genetex, Irvine, CA, USA, GTX-100446), and horseradish peroxidase (HRP)-conjugated secondary antibody (Jackson ImmunoResearch laboratories, Inc., West grove, PA, USA).

### 4.3. 3-(4,5-dimethylthiazol-2-yl)-2,5-diphenyltetrazolium bromide (MTT) Assay

Three thousand cells were seeded in every well of a 96-well plate. The cells were treated with the indicated concentrations of SGC8158 for 2 days, followed by treatment with 0.5 mg/mL MTT for an additional 3 h. The medium was then removed, and 120 μL of dimethyl sulfoxide was added to each well. Absorbance was measured at 590 nm using an Epoch microplate spectrophotometer (Biotek, Winooski, VT, USA). The IC_50_ values were calculated using the GraphPad Prism version 8.4 (GraphPad, San Diego, CA, USA).

### 4.4. Colony Forming Assay

Cells were treated with 10 μM SGC8158 or transfected with PRMT7 siRNA for three days, followed by supplementation with fresh complete media. After 5 days of culture, the colonies were fixed with 4% paraformaldehyde (PFA) and stained using 0.05% crystal violet. The stained cells were washed thrice with deionized water and then dried. After dissolving the stained crystal violet in a 30% acetic acid solution, the absorbance at 540 nm was measured using an Epoch microplate spectrophotometer.

### 4.5. Cell Cycle Analysis

To analyze cell cycle progression, A549 cells were harvested using trypsin. The cells were rinsed with cold phosphate-buffered saline (PBS) and fixed with 70% ethanol for at least 1 h on ice. After fixation, the cells were centrifuged at 3000 rpm for 3 min, and the supernatant was removed. After washing with cold PBS, the cells were resuspended in 0.25 mL PBS and RNase A (Thermo Fisher Scientific, Waltham, MA, USA), and then 10 μg/mL propidium iodide (BD Biosciences, Franklin Lakes, NJ, USA) was added. Data were measured using a FACSCalibur (BD Biosciences) and analyzed using FlowJo software.

### 4.6. Western Blot

Cells were lysed using NP-40 lysis buffer (10 mM Tris HCl pH 7.4, 100 mM NaCl, 1 mM ethylenediaminetetraacetic acid (EDTA), 1 mM ethylene glycol-bis(2-aminoethylether)-N,N,N′,N′-tetraacetic acid (EGTA), 1% NP-40, and 10% glycerol) and radioimmunoprecipitation assay (RIPA) lysis buffer (150 mM NaCl, 50 mM Tris HCl pH 8.0, 1% NP-40, 0.5% sodium deoxycholate, and 0.1% sodium dodecyl sulfate (SDS)) supplemented with 1 mM 1,4-dithiothreitol (DTT; Roche, Basel, Switzerland) and 1× protease and phosphatase inhibitor cocktails (Roche). The sonicated cell lysates were centrifuged at 13,000 rpm for 10 min. Protein levels were quantified by Bradford assay using bovine serum albumin (BSA) solution (Thermo Fisher Scientific) according to the manufacturer’s instructions. Equal amounts of protein were subjected to SDS-polyacrylamide gel electrophoresis (SDS-PAGE) and transferred onto a polyvinylidene difluoride (PVDF) membrane (Millipore, Billerica, MA, USA). The membranes were blocked with 5% skimmed milk in Tris-buffered saline (TBS) containing 0.1% Tween-20 (TBS-T) for at least 1 h at room temperature (RT). After washing, the membranes were incubated with primary antibodies at 4 °C overnight. After washing three times with TBS-T, the membranes were incubated with an HRP-conjugated secondary antibody at RT for 1 h. Signals were detected using a chemiluminescent detection reagent (Advansta, Inc., San Jose, CA, USA). The bands were measured using the ImageJ software (https://imagej.nih.gov/ij/, accessed on 12 August 2022).

### 4.7. Quantitative Reverse-Transcription PCR

Total cellular RNA was extracted using TRIsure^TM^ (Bioline, London, UK) and cDNA was synthesized using the SensiFAST^TM^ cDNA kit (Bioline). The synthesized cDNA was subjected to qPCR amplification using SensiFAST SYBRTM No-ROX premix (Bioline) and Eco Real-time PCR ver 3.1 (Illumina, San Diego, CA, USA). The reaction parameters were as follows: cDNA synthesis at 40 °C for 1 h, transcriptase inactivation at 85 °C for 5 min, and 4 °C for 5 min. PCR was performed at 95 °C for 10 s, 58 °C for 20 s, and 72 °C for 20 s for 40 cycles. The primer sets used were as follows: 28S ribosomal RNA, forward: 5′-AACGAGATTCCCACTGTCCC-3′, reverse: 3′-TTGCTCTAAGGGTGACAGGG-5′; and p21, forward: 5′-TGTCACTGTCTTGTACCCTTG-3′, reverse: 3′-ACAGTGACAGAACATGGGAAC-5′.

### 4.8. Immunoprecipitation

Whole-cell extracts were obtained using NP-40 lysis buffer (10 mM Tris-HCl pH 7.4, 100 mM NaCl, 1 mM EDTA, 1 mM EGTA, 1% NP-40, and 10% glycerol) supplemented with 1 mM DTT (Roche) and 1× protease inhibitor cocktail (Roche). One milligram protein of cell lysates was used for immunoprecipitation assay, and appropriate antibodies (1 μg) were added and incubated overnight at 4 °C on a rotator, followed by antibody-protein complex capture with protein A/G sepharose beads (Santa Cruz Biotechnology) for at least 2 h at 4 °C. After washing three times with NP-40 lysis buffer, the complexes were eluted and analyzed using SDS-PAGE and immunoblotting.

### 4.9. Senescence-Associated β-Galactosidase Assay

Cells were fixed with 4% PFA for 10 min at RT, washed, and incubated overnight with β-galactosidase staining solution at 37 °C without CO_2_. This procedure was performed using a β-galactosidase staining kit (Cell Signaling Technology), according to the manufacturer’s protocol. After obtaining a blue color, the cells were observed under a light microscope (Nikon Eclipse TS 100, Tokyo, Japan).

### 4.10. Immunofluorescence and Confocal Microscopy 

Cells were fixed with 4% PFA for 15 min, washed three times with cold PBS, and permeabilized with 0.5% Triton X-100 for 15 min at RT. After overnight incubation with each primary antibody at 4 °C, the cells were washed with cold PBS, followed by incubation with fluorescence-conjugated secondary antibody for 1 h at RT in the dark. After staining with 4′,6-diamidino-2-phenylindole (DAPI) (Thermo Fisher Scientific) for 5 min, the cells were mounted onto glass slides. The stained cells were visualized using a Zeiss LSM 710 confocal microscope (Carl Zeiss, Oberkochen, Germany).

### 4.11. HR and NHEJ Assay

U2OS stable cell lines harboring DR-GFP or Ej5-GFP plasmids were kindly gifted by Prof. Yonghwan Kim (Sookmyung Women’s University, Seoul, Korea). DR-GFP- or Ej5-GFP-harboring cells were seeded in 6-well plates and co-transfected with 1 μg of I-SceI vector with negative control siRNA or treated compound using Transit-X2^TM^ (Mirus Bio). After 2 days, the cells were harvested using trypsin and washed with PBS. GFP signaling arising from recombination events was measured using flow cytometry (FACSCalibur, BD Bioscience). Fluorescence was detected in the FL1-H channel (logarithmic scale).

### 4.12. CI and Data Processing

To investigate the synergistic effects of SGC8158 and doxorubicin, cell viability was evaluated by an MTT assay using a combination of the two drugs. The synergistic effects were determined using the CI and Bliss synergy score. The CI values were calculated using the CompuSyn software (ComboSyn, Inc., Paramus, NJ, USA). This software drew a plot using the following formula: CI = (D)1/(Dx)1 + (D)2/(Dx)2. (Dx)1 and (Dx)2 are the concentration of a single drug that is effective by X%, and (D)1 and (D)2 are the drug concentrations of a combination treatment that are effective by X%. When the CI value is less than 1, a synergistic effect is indicated, whereas when it is 1 or greater, it is an additive or antagonistic effect. Bliss synergy scores were calculated using the Bliss independence model in Synergyfinder (https://synergyfinder.fimm.fi/, accessed on 30 August 2022). A Bliss synergy score greater than five was considered to represent synergism. 

## Figures and Tables

**Figure 1 ijms-23-12323-f001:**
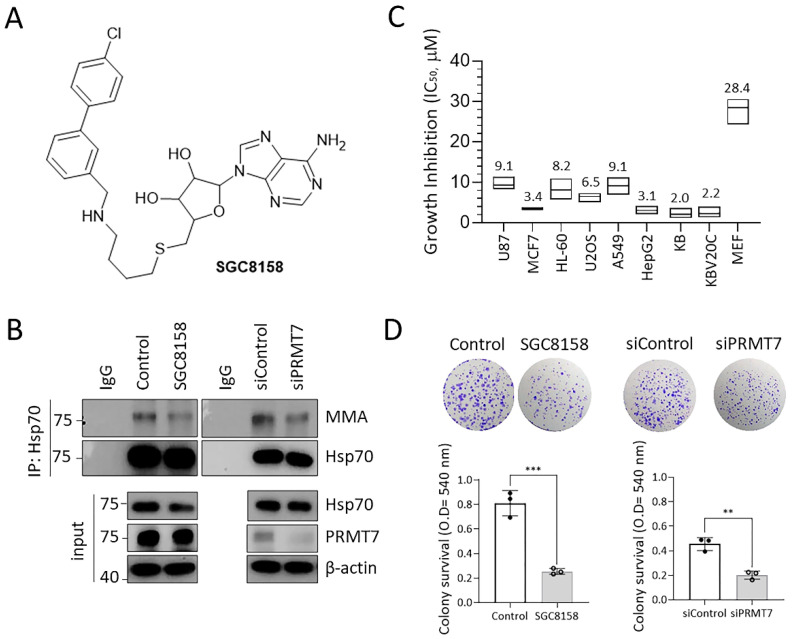
PRMT7 inhibition causes growth inhibition in various human cancer cells. (**A**) Structure of SGC8158 (**B**) A549 cells were treated with 10 μM SGC8158 or transfected with PRMT7 siRNA for 3 days. Whole-cell lysates were subjected to immunoprecipitation using Hsp70 antibody, followed by immunoblotting with MMA antibodies. (**C**) The growth inhibition of SGC8158 in various human cancer cells was determined using an MTT assay. Data are presented as floating bar plots from three independent experiments. (**D**) A549 cells were treated with 10 μM SGC8158 or transfected with PRMT7 siRNA as described. After five days, the colonies were stained using crystal violet (10× magnification) and the absorbance was measured at 540 nm. Data are presented as means ± SD (*n* = 3). ** *p* < 0.01 and *** *p* < 0.001. MMA: monomethylarginine; MTT: 3-(4,5-dimethylthiazol-2-yl)-2,5-diphenyltetrazolium bromide; SD: standard deviation.

**Figure 2 ijms-23-12323-f002:**
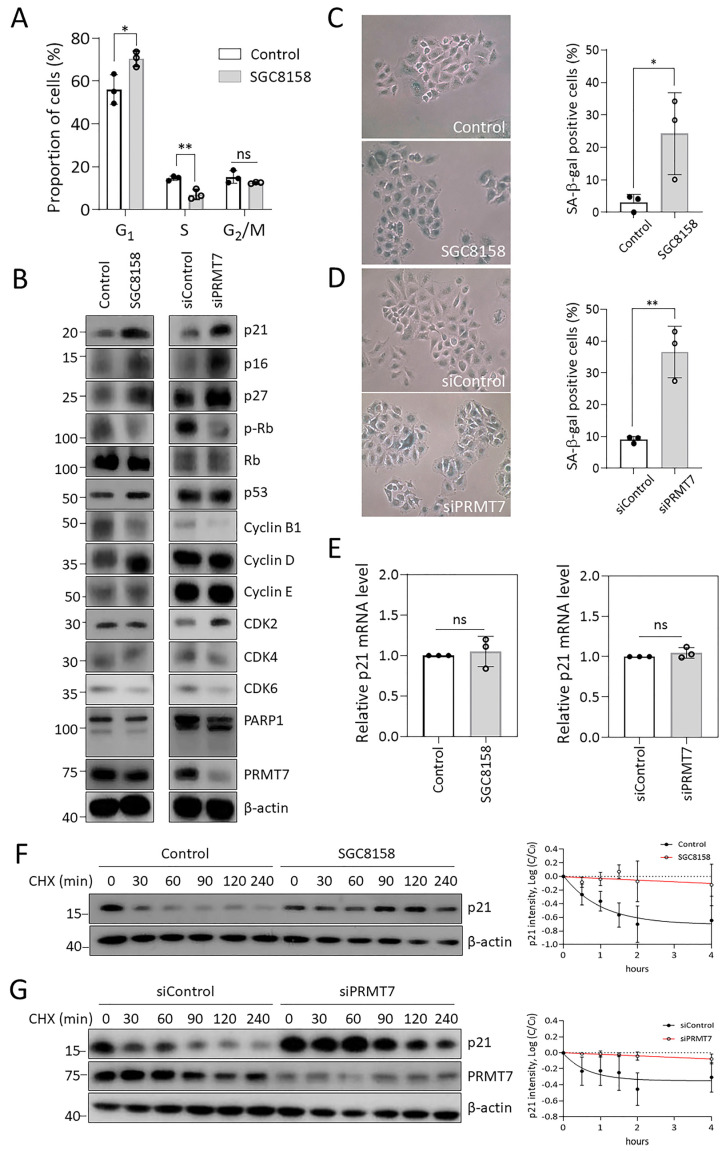
PRMT7 inhibition induces cell cycle arrest and cellular senescence via p21 accumulation. (**A**) After treatment of A549 cells with 10 μM SGC8158 for 3 days, the proportion of cells in each phase of the cell cycle was measured by FACS analysis. Data are presented as means ± SD of three independent experiments. * *p* < 0.05 and ** *p* < 0.01. (**B**) The expression profiles of cell cycle-related proteins. After treatment of A549 cells with 10 μM SGC8158 or PRMT7 siRNA transfection for 3 days, the levels of cell cycle-related proteins were analyzed by western blotting. (**C**,**D**) Representative images illustrating SA-β-gal-positive cells (100× magnification). Percentage of SA-β-gal-positive cells was shown as means ± SD from three independent experiments. * *p* < 0.05 and ** *p* < 0.01. (**E**) p21 mRNA levels were evaluated by quantitative RT-PCR and presented as means ± SD (*n* = 3). (**F**,**G**) The cells were treated with 10 μM SGC8158 or transfected with PRMT7 siRNA for 3 days, and then treated with 100 μg/mL CHX for the indicated times. Band intensities were quantitated using image processing software. Error bars indicate the SD of three independent replicates. FACS: fluorescent-activated cell sorting; SD: standard deviation; CHX: cycloheximide.

**Figure 3 ijms-23-12323-f003:**
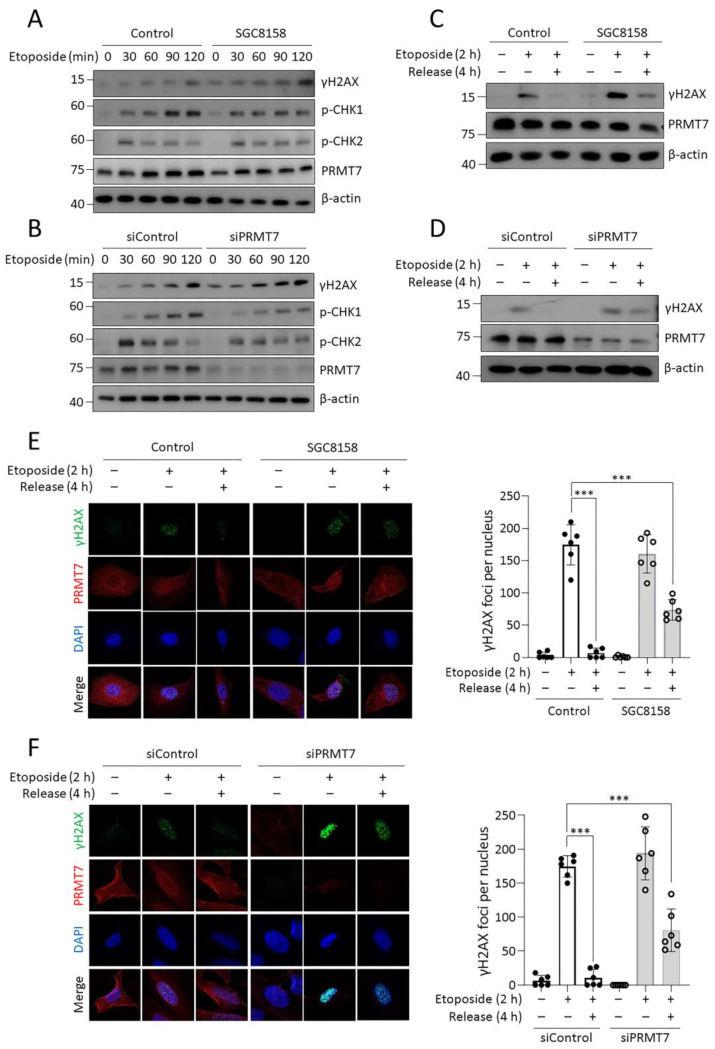
PRMT7 regulates DNA damage response. (**A**,**B**) Levels of DNA damage-associated proteins using western blotting. U2OS cells were treated with 10 μM SGC8158 (**A**) or transfected with PRMT7 siRNA (**B**) for 2 days, followed by treatment with 10 μM etoposide for indicated times. Cell lysates were subjected to immunoblotting analysis with the indicated antibodies. (**C**–**F**) Analysis of γH2AX levels during DNA repair process. After suppressing PRMT7 in U2OS cells by treating with 10 μM SGC8158 (**C**,**E**) or the siRNA (**D**,**F**) for 2 days, the cells were treated with 10 μM etoposide for 2 h, and then further incubated with fresh medium without etoposide for an additional 4 h. γH2AX levels were measured using western blotting (**C**,**D**) or immunostaining (γH2AX; green, PRMT7; red, DAPI; blue) (**E**,**F**). The fluorescence intensity of γH2AX was quantified by image analysis software (630× magnification). Data are presented as means ± SD of six independent experiments. *** *p* < 0.001. DAPI: 4′,6-diamidino-2-phenylindole; SD: standard deviation.

**Figure 4 ijms-23-12323-f004:**
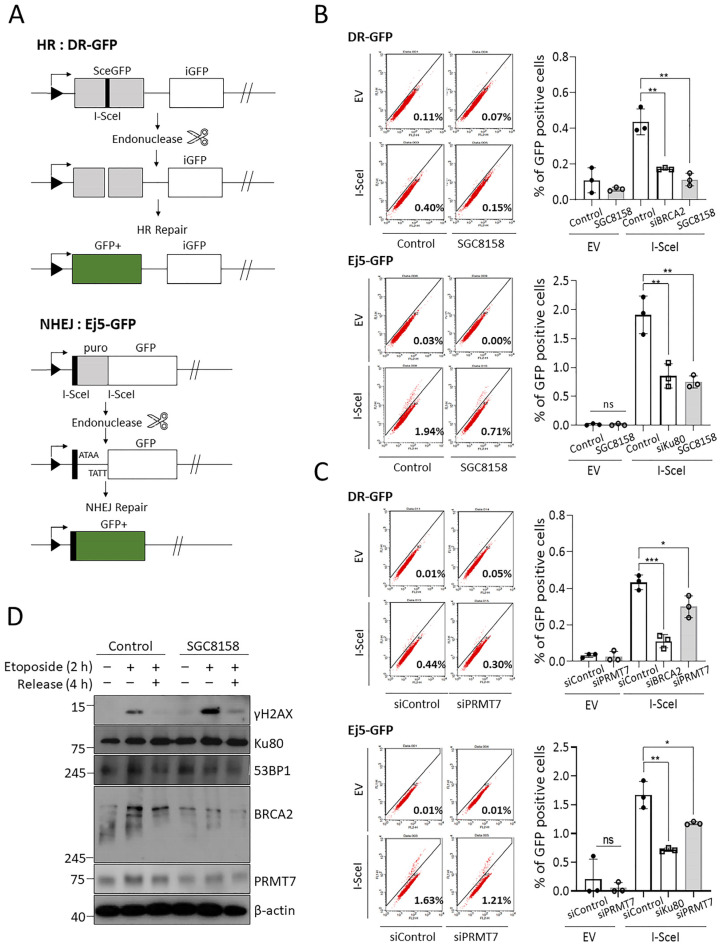
PRMT7 regulates DNA repair processes. Schematic of the DR-GFP and Ej5-GFP reporter assays. (**A**) DR-GFP reporter system was used to monitor HR process. Modified GFP gene (*SceGFP*) contains an I-SceI site in-frame. Expression of I-SceI causes DSBs at internal *SceGFP* sites and a non-functional GFP fragment gene (*iGFP*) is used to repair the DSB by HR, generating a functional *GFP* gene. In addition, the Ej5-GFP reporter system was used to monitor the total NHEJ process. The *GFP* gene is separated from its promoter by a puromycin resistance gene (*puro*) flanked by two I-SceI sites. Transient expression of I-SceI endonuclease causes the excision of the *puro* and induces the repair of the DSB by NHEJ, thus creating a functional *GFP* gene. (**B**,**C**) U2OS stable cell lines expressing DR-GFP or Ej5-GFP reporter constructs were co-treated (or transfected) with PRMT7 inhibitor (or siRNA) and I-SceI plasmid. After 3 days, the efficiency of HR or NHEJ was determined by counting the number of GFP-positive cells using flow cytometry. *BRCA2* siRNA and *Ku80* siRNA were used as positive controls to validate HR and NHEJ reporter systems, respectively. The percentage of GFP-positive cells from three independent experiments is indicated as the means ± SD. * *p* < 0.05, ** *p* < 0.01, and *** *p* < 0.001. (**D**) The expression levels of HR or NHEJ-related proteins (BRCA2, Ku80, and 53BP1) in the presence of SGC8158 were determined using western blotting under the same conditions. HR: homologous recombination; DSB: double-strand break; NHEJ: non-homologous end-joining; SD: standard deviation.

**Figure 5 ijms-23-12323-f005:**
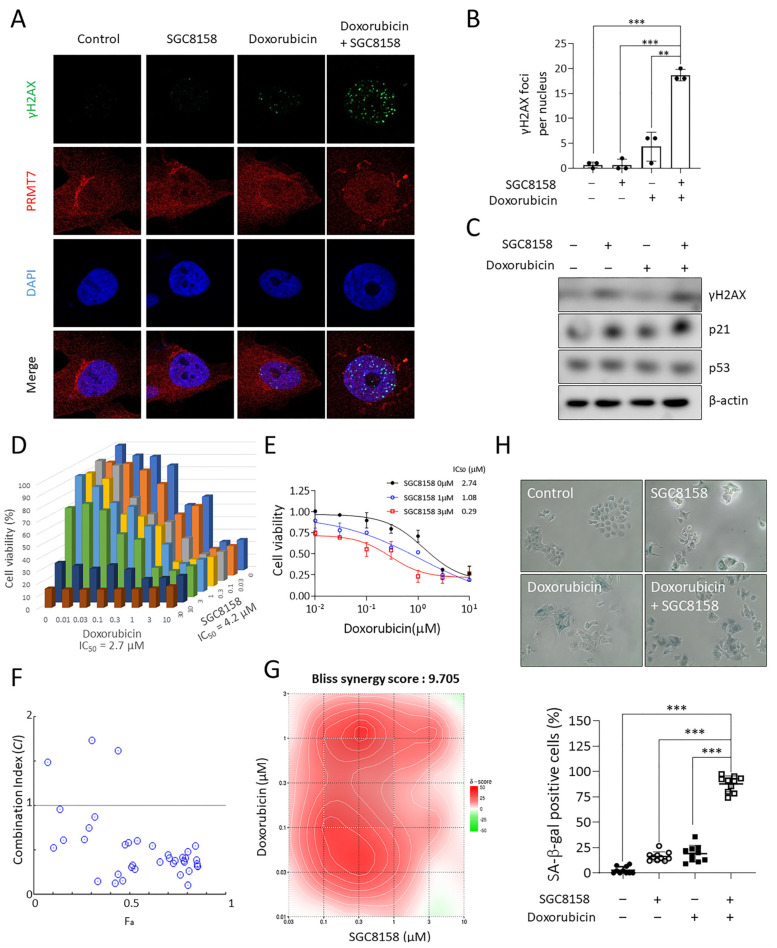
SGC8158 potentiates doxorubicin-induced DNA damage response and its cytotoxicity. After 2 days treatment of MCF7 cells with the combination of 0.01 μM doxorubicin and 5 μM SGC8158, (**A**) γH2AX foci were visualized using confocal microscopy (630× magnification) and (**B**) quantification of cells showing high fluorescence of γH2AX was performed using image processing software (*n* = 3). ** *p* < 0.01 and *** *p* < 0.001. (**C**) In the same conditions, the γH2AX levels were measured using western blotting. (**D**–**G**) MCF7 cells were treated with a combinatorial setting of SGC8158 and doxorubicin. (**D**) Cell viabilities were evaluated by an MTT assay. (**E**) The synergistic effects were assessed by comparing the IC_50_ of doxorubicin in the presence of SGC8158 (1 or 3 μM). (**F**) The CI values were calculated using CompuSyn software. (**G**) The Bliss synergy score was calculated using the Bliss independence model. The gradation of the red regions indicates the intensity of synergism. (**H**) Representative images illustrating SA-β-gal-positive cells under the experimental setting (A) (100× magnification). Percentage of SA-β-gal-positive cells was shown as means ± SD (*n* = 10). *** *p* < 0.001. MTT: 3-(4,5-dimethylthiazol-2-yl)-2,5-diphenyltetrazolium bromide; CI: combination index; SD: standard deviation.

## Data Availability

Not applicable.

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
