# Peer review of "PRMT7 Inhibitor SGC8158 Enhances Doxorubicin-Induced DNA Damage and Its Cytotoxicity"

_ijms, 2022, doi:10.3390/ijms232012323_

Round 1

Reviewer 1 Report

In this manuscript Jeong and collaborators are showing that the methyltransferase PRMT7 can sensitize cancer cells to chemotherapeutics. Mechanistically authors provide data to show that DNA break repair is defective when PRMT7 is depleted or inhibited in cancer cells.

Altogether the paper is very well written and contains all details to be understood and experiments be repeated. Next, the data are collectively strong, clearly presented, and well-interpreted. According to the importance of finding new targets in cancer treatment I recommend the publications of these important results. Here below, I list a few minor points that, I hope, will help authors to finalize the paper before its publication.

Minor points.

1.     In introduction, it is stated that ‘PRMT7 is expressed in various cancers’ (references 10 and 11). Can authors be more precise? Is it expressed of overexpressed? What is the expression in normal cells.

2.     Figure 1C. Has the MTT assay been performed in 3 independent experiments?

3.     Figure 1C. To my view, the result would be more convincing if non-cancerous cell lines would be included. The question is to know whether only cancer cells are sensitive to PRMT7 inhibitor?

4.     Figure 3A and 3B. The blots are not completely convincing compared to immunofluorescence and DNA break repair assays. Also, it raises a question. If DNA break repair is not fully functional, one would expect not only to delay DNA damage signaling but also to increase the level of the signaling as unrepaired breaks would accumulate during etoposide treatment (I do see this increase gH2AX in figure 3C).

5.     Figure 4D. Is it the same blot as figure 3C? If yes, it has to be stated in the figure legends.

Reviewer 2 Report

The manuscript (ijms-1957637) provided by Jeong et al. presents research involving PRMT7 in the context of its role in cell cycle progression and response to DNA damage in cancer cells. To inhibit PRMT7 authors used potent and selective PRMT7 inhibitor- SGC8158 and siRNA transfection. It should be regarded as a great advantage of this research. This study revealed that PRMT7 is associated with DNA repair and that combination of SGC8158 with doxorubicin causes greater DNA damage and cytotoxicity on cancer cells. This research is very detailed and innovative, the results of this work might be of interest to the readers of Molecules. However, the work needs to be improved on the following points:

 Title

 - It seems to the reviewer that the title of the paper is a bit poorly worded and therefore it can be misunderstood. The current title shows that doxorubicin induces DNA damage response, i.e. that doxorubicin induces DNA repair, which is the complete opposite of its action. The title should be corrected to indicate that SGC8158 enhances DOX-induced DNA damage and enhances its cytotoxicity, not DNA repair.

 Abstract

- In the case of the synergistic effect of SGC8158 and doxorubicin the types of cancer and name of cancer cell lines should be specified in the abstract.

Introduction

- The entire introduction is too detailed and too extensive. It should be shortened by focusing on the proven functions of PRMT7 and its relationship to DDR but without much detail.

- „PRMT7 is expressed in various cancers, including breast, lung, bone, and liver cancers”
In cited articles (10,11) reviewer do not find information about PRMT7 expression in lung, bone, and liver cancer. Supplement the introduction with relevant citations.

Results

Subsection 2.1

- The phrase "growth retardation" is incorrect, please change it to growth inhibition or rewrite the sentence. – In the title of the 2.1 Subsection.  

- “Recently, SGC8158 (Figure 1A), an active form of SGC3027, was identified as a potent and competitive inhibitor of PRMT7”- The sentence repeats in "Introduction" and subsection 2.1 in the "Results" section. Authors should consider leaving this sentence in the "Introduction" section and moving Figure 1A as a separate figure to "Introduction"

- Figure 1B and other Figures- Explain the abbreviation "CON" or correct the figures with the full word "Control"

- „Since MDR has been considered a major hurdle for successful cancer therapy [30,31], it is noteworthy that SGC8158 is also effective in MDR cancer cells, providing an important clue in developing strategies to overcome MDR.” - This sentence should be included in the discussion rather than in the description of the results.- Subsection 2.1

 -  A non-cancer cell line should be included in the MTT assay to test the toxicity of the SGC8158.

- The reviewer's major doubts are raised by the differences in incubation times of the tested cells with SGC8158 in various tests. In the MTT assay (Figure 1C) it was 48h but in the other tests like colony formation assay, PRMT7 inhibition assay (Figure 1B), and cell cycle analysis it was 72h. And again in the test of a combination of SGC8158 with doxorubicin, the incubation time was 48 h. What are the reasons for the differences in incubation times?

Subsection 2.2

- The reviewer is not convinced that the 14% increase in the G1 phase compared to the control is biologically significant. In other publications, cell cycle arrest in the G1 phase is determined by an increase in the G1 phase of 30% and more.

- The photos shown in Figures 2C and 2D are too small and of too low quality. Blue staining is hardly noticeable.

- The unit of incubation time (days or hours) should be standardized throughout the manuscript.

Subsection 2.5

- It is necessary to supplement the information in the description of the results in subsection 2.5 about the tested cell line that it was MCF-7.

- There is no explanation in the manuscript why the breast cancer cell line (MCF-7) and osteosarcoma cell line (U2OS) were used in the SGC8158 and doxorubicin combination test. Why these types of cancers?

-  Figure 5D - not very clear way of presenting the results. It would be better to select and represent those several different concentrations of compounds at which a synergistic effect is present. Additionally, there is no legend regarding the used colors.

Materials and methods

Subsection 4.1

- “A549 (human non-small lung cancer cells), KB (human oral squamous cell carcinoma cells), and HL-60 (human acute myeloid leukemia cells) were cultured in RPMI1640 (Roswell Park Memorial Institute-1640) supplemented with FBS and 100 U/mL penicillin-streptomycin (Hyclone Laboratories).How many % FBS? – Subsection 4.1

- “Multidrug-resistant KBV20C cells were derived from KB cells and cultured with 20 nM vincristine under growth conditions to preserve MDR characteristics.” Is there any research paper proving that this cell line should be cultured this way?- Subsection 4.1

Subsection 4.3

- How many cells were seeded in 96-well plates? (cells/well or cells/mL)

- How was the IC50 calculated?

Subsection 4.7

- Primer sequences can be present in the form of the table to make them more readable. 

Kind regards

Round 2

Reviewer 2 Report

The reviewer accepts the manuscript after these major revisions.